# TOWARD A UNIFIED FRAMEWORK FOR DATA-EFFICIENT EVALUATION OF LARGE LANGUAGE MODELS

## ABSTRACT

Evaluating large language models (LLMs) on comprehensive benchmarks is a cornerstone of their development, yet it's often computationally and financially prohibitive. While Item Response Theory (IRT) offers a promising path toward data-efficient evaluation by disentangling model capability from item difficulty, existing IRT-based methods are hampered by significant limitations. They are typically restricted to binary correctness metrics, failing to natively handle the continuous scores used in generative tasks, and they operate on single benchmarks, ignoring valuable structural knowledge like correlations across different metrics or benchmarks. To overcome these challenges, we introduce LEGO-IRT, a unified and flexible framework for data-efficient LLM evaluation. LEGO-IRT's novel design natively supports both binary and continuous evaluation metrics. Moreover, it introduces a factorized architecture to explicitly model and leverage structural knowledge, decomposing model ability estimates into a general component and structure-specific (e.g., per-metric or per-benchmark) components. Through extensive experiments involving 70 LLMs across 5 benchmarks, we show that LEGO-IRT achieves stable capability estimates using just $3\%$ of the total evaluation items. We demonstrate that incorporating structural knowledge reduces estimation error by up to $10\%$ and reveal that the latent abilities estimated by our framework may align more closely with human preferences.

## 1 INTRODUCTION

The evaluation of model performance is a critical component in the development process of modern large language models (LLMs) as well as a building block towards deeper understandings of LLMs. Currently, the prevailing practice for LLM evaluation is to leverage existing benchmarks (Hendrycks et al., 2021; Liang et al., 2022; Chang et al., 2024) by first running an inference procedure over all the evaluation items in the benchmark, followed by a grading step that produces decisions about the performance of the candidate model over individual benchmark items. The final result is then computed via simply averaging over individual performance measures, with the majority being a binary judgment indicating correctness. Despite its simplicity, conducting a thorough evaluation procedure like HELM (Liang et al., 2022) invorlves hundreds of thousands of items, making the inferential cost substantial, sometimes prohibitive, both computationally (thousands of GPU hours) and financially (inference cost for state-of-the-art proprietary models (Jaech et al., 2024; Comanici et al., 2025)). The challenge is even more evident under the surging development of thinking-style models that exploit inference-time scaling (Guo et al., 2025), where the models may require a significant amount of tokens before arriving at the final answer.

Effort has been made toward data-efficient evaluation of LLMs by utilizing subsets of the (full) benchmarks (Liang et al., 2022; Vivek et al., 2023; Saranathan et al., 2024; 2025). However, directly comparing models using average scores between (random) subset evaluations is unreliable (Truong et al., 2025), as the problems' difficulty may serve as a confounding factor. To alleviate this issue, recent developments (Polo et al., 2024; Truong et al., 2025; Zhou et al., 2025) utilize ideas from item response theory (IRT) (Chen et al., 2025) to produce robust performance estimations that are stable across subsets. In a nutshell, IRT approaches disentangle the influences of model capability

| Method | Stability | Binary metric | Continuous metric | Multiple metrics | Multiple benchmarks |
|---|---|---|---|---|---|
| Mean aggregation | low | ✓ | ✓ | ✗ | ✓ |
| Polo et al. (2024) | high | ✓ | ✓ | ✗ | ✗ |
| Truong et al. (2025) | high | ✓ | ✗ | ✗ | ✗ |
| **LEGO-IRT** | high | ✓ | ✓ | ✓ | ✓ |

Table 1: A comparison among contemporary methods on (data-efficient) LLM evaluation.

and evaluation item difficulty, thereby producing *invariant estimates of model capabilities*. Despite its theoretical advantage, contemporary IRT-based solutions still face multiple challenges:

**Limited applicability** As IRT-based paradigms are *model-based*, they rely on probabilistic assumptions over the evaluation metric, which is either restricted to be binary (Truong et al., 2025) or binarized from continuous metrics (Polo et al., 2024). As LLMs are inherently generative models, binary metrics can only serve as a grading mechanism for the correctness of the final answer, but fail to provide fine-grained evaluation of model generations (Lightman et al., 2023). Moreover, for sequence-to-sequence (seq2seq) tasks like machine translation or article summarization, the conventional practice is to use *continuous metrics* such as BLEU (Papineni et al., 2002), ROUGE (Lin, 2004) or BERTScore (Zhang et al., 2019).

**Lack of structural knowledge** Previous developments on IRT for LLM evaluation operate on a *single metric, single benchmark* scheme. For evaluation tasks where multiple metrics are applicable, it is often the case that distinct metrics may capture different characteristics of model performance. For example, in seq2seq task evaluation, BLEU emphasizes precision, ROUGE focuses on recall, while BERTScore incorporates semantic similarity. Simultaneously evaluating model performance under multiple metrics suggests a potential improvement in modeling efficacy through properly designed joint modeling across different metrics. Furthermore, it has been widely recognized that model performances exhibit a certain sense of correlation among benchmarks (Perlitz et al., 2024b). It is therefore worthwhile to investigate whether model-based LLM evaluations can further benefit from a better exploitation of *structural knowledge* like inter-metric or inter-benchmark correlation that goes beyond modeling over a fixed metric over a single benchmark. In this paper, we address the aforementioned challenges by developing a flexible and unified framework for applying IRT-based modeling frameworks to the evaluation of LLMs, which we termed **L**anguage model **E**valuation under **G**eneral **O**utcomes based on **I**tem **R**esponse **T**heory (LEGO-IRT), with the following summarized contributions:

**Flexible metric types** With a novel IRT model design, LEGO-IRT supports modeling both binary metrics as well as continuous ones in a *native* fashion, i.e., no discretization is required.

**Structural knowledge injection through factorization** Inspired by recent developments in IRT (Fang et al., 2021), LEGO-IRT extends contemporary IRT frameworks by introducing factorized designs that effectively decompose model ability estimates into a general component plus structure-specific offsets that properly handle scenarios where multiple metrics or multiple benchmarks are involved.

**Empirical validation** Through extensive empirical investigation involving 70 latest state-of-the-art LLMs as well as 5 benchmarks comprising distinct evaluation types, LEGO-IRT is demonstrated to obtain stable estimates of model capabilities while requiring only 3% of total items. Moreover, we validate the advantage of incorporating structural knowledge by showing up to 10% error reduction in model performance estimation. We also reveal interesting findings that the latent model abilities estimated through LEGO-IRT might align better with human preferences.

## 2 RELATED WORKS

### 2.1 DATA-EFFICIENT EVALUATION OF LARGE LANGUAGE MODELS

The increasing versatility of LLMs has given rise to holistic evaluation benchmarks such as HELM (Liang et al., 2022) that comprehensively assess a broad range of model capabilities but requires considerable expenditure [1]. Data-efficient evaluation methods have recently emerged (Perlitz et al., 2024a; Vivek et al., 2023; Saranathan et al., 2025; Li et al., 2025) that aim at reducing the evaluation cost via shrinking the size of the benchmarks, utilizing techniques such as adaptive sampling (Xu et al., 2024; Saranathan et al., 2025), coreset identification (Vivek et al., 2023), and active learning (Li et al., 2025). While these subset-based approaches have empirically shown to achieve reasonable

---

[1] As reported in the HELM paper (Liang et al., 2022), it cost $38001 for commercial api and approximately 19500 GPU hours' compute to evaluate just 30 LLMs across 13 distinct tasks.

evaluation quality, they are typically derived from experimental observations or heuristics, lacking a principled methodological foundation.

## 2.2 ITEM RESPONSE THEORY (IRT) AND LANGUAGE MODEL EVALUATION

The strong theoretical foundations of IRT (Chen et al., 2025) have inspired novel paradigms in language model evaluation (Lalor et al., 2016; Zhuang et al., 2025). Polo et al. (2024) used IRT methods to reduce the evaluation effort over MMLU (Hendrycks et al., 2021) by $99\%$ while leaving the evaluation quality almost intact. A recent study Truong et al. (2025) pushed IRT-based LLM evaluation to HELM scale, and proposed efficient algorithms to speed up IRT modeling. The application of IRT is also proliferating to other LLM-related tasks, including applications in RAG pipeline design (Guinet et al., 2024) and improving arena-type LLM comparisons (Liu et al., 2025). A notable property of the IRT paradigm is that it allows *model-based evaluations* that could be potentially facilitated to predict the performance of LLMs on unseen benchmarks *without* running the actual inference, which is closely connected to the field of unsupervised risk estimation (Donmez et al., 2010; Platanios et al., 2016).

## 3 WARMUP: ITEM RESPONSE THEORY(IRT)

Item response theory (IRT) models, also referred to as latent trait models, play an important role in educational testing and psychological measurement as well as several other areas of behavioral and cognitive measurement (Chen et al., 2025). In psychometrics, IRT has been proven to be one of the most fundamental tools in the construction, evaluation, and scoring of large-scale high-stakes educational tests (Birdsall, 2011; Robin et al., 2014). IRT models are, in fact, probabilistic models for individuals' responses to a set of items, where the responses are typically binary. These models are latent variable models from a statistical perspective, dating back to Spearman's factor model for intelligence. Rasch models, introduced in 1960s (Rasch, 1960), laid the foundation of IRT as a theory for educational and psychological testing. Later on, the two-parameter (2PL) and three-parameter logistic (3PL) models (Birnbaum, 1968) were developed that are still widely used in educational testing these days.

The Rasch model assumes local independence, and it postulates the following form,

$$\Pr(Y_{ij} = 1 \mid \theta_i, b_j) = \sigma(\theta_i - b_j), \quad \sigma(x) = \frac{1}{1 + e^{-x}}, \tag{1}$$

where $Y_{ij} \in \{0, 1\}$ indicates whether $i$-th LLM correctly responsd item $j$; $\theta_i$ is treated as the $i$-th LLM's latent ability and $b_j$ is viewed as the difficulty of item $j$. In the context of LLM evaluation, an item stands for a problem or question that belongs to some benchmark. Hereafter, we will refer to $Y_{ij}$ as either *response* or *metric* interchangeably with meanings clear from the context. Similarly, 2PL and 3PL have the following model structures,

$$\Pr(Y_{ij} = 1 \mid \theta_i, b_j) = \sigma(a_j \theta_i - b_j) \tag{2}$$

and

$$\Pr(Y_{ij} = 1 \mid \theta_i, b_j) = c_j + (1 - c_j)\sigma(\theta_i - b_j), \tag{3}$$

where $a_j$ is known as the discriminative parameter and guess parameter of item $j$.

Given a specified model, evaluation typically proceeds in two stages, *calibration* and *scoring*.

**Calibration stage** In this stage, the test developer collects a response matrix $Y \in \{0, 1\}^{N \times J}$, where $N$ and $J$ denote the number of LLMs and items, respectively. Each $Y_{ij}$ indicates the response of LLM $i$ to item $j$. The LLM-specific parameters $\theta_i$'s and item-specific parameters are then jointly estimated. The outcome of calibration is a set of calibrated items with estimated difficulties $\{b_j\}$ and associated uncertainty quantifications, as well as the estimated latent abilities $\{\theta_i\}$ of LLMs with their credible intervals.

**Scoring stage** In this stage, the abilities of new LLMs are estimated while keeping the item parameters fixed at the calibrated values. Note that with fixed item parameters, the corresponding estimation problem is reduced to simple logistic regression, which is extremely fast to compute.

## 4 THE LEGO-IRT FRAMEWORK

### 4.1 ADAPTATION TO CONTINUOUS METRICS

Under LLM evaluation scenarios such as seq2seq, most of the widely adopted metrics, such as BLEU, ROUGE, or BERTScore, take values in $[0, 1]$. However, the classical IRT model only handles $\{0, 1\}$ correctness and is unable to capture these continuous variations. To address this, we construct a novel continuous IRT model that effectively accommodates continuous responses, which serves as the first component in our proposed LEGO-IRT framework termed LEGO-CM (CM as abbreviation of **C**ontinuous **M**etric). Suppose there are $N$ LLMs and $J$ items. Let $Y_{ij} \in (0, 1)$ denote the score of $i$-th LLM on $j$-th item. The distribution of $Y_{ij}$ under LEGO-CM postulates the following form,

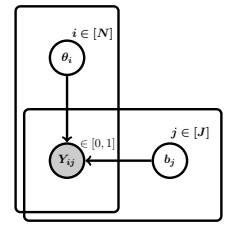

Figure 1: PGM description of LEGO-CM

$$\text{logit}(Y_{ij}) \sim \mathcal{N}(a_j \theta_i - b_j, \sigma_j^2). \qquad \text{(LEGO-CM)}$$

where $\text{logit}(y) = \log \frac{y}{1-y}$, $\theta_i$ denotes the latent ability of LLM $i$, $a_j > 0$ is the discrimination parameter, $b_j$ is the difficulty parameter, and $\sigma_j > 0$ measures item-specific dispersion or scoring noise. The density function can be written as

$$p(y_{ij} \mid \theta_i, a_j, b_j, \sigma_j) = \frac{1}{\sqrt{2\pi}\sigma_j} \exp\left\{ -\frac{[\text{logit}(y_{ij}) - (a_j\theta_i - b_j)]^2}{2\sigma_j^2} \right\}. \qquad (4)$$

We present a probabilistic graphical model (PGM) description of LEGO-CM in figure 1.

**Parameter estimation** In this paper, we take a Bayesian approach to the estimation procedure across all LEGO-IRT using Markov chain Monte Carlo (MCMC; (Metropolis et al., 1953)). While previous works have adopted alternative approaches like expectation-maximization (EM) (Truong et al., 2025), we found EM to be less flexible than MCMC to apply in a unified modeling context. A more detailed discussion between EM and MCMC is postponed to appendix D. To implement the MCMC for LEGO-CM, we choose the following priors,

$$\theta_i \sim \mathcal{N}(0, 1), \quad a_j \sim \text{LogNormal}(\mu_a, \sigma_a^2), \quad b_j \sim \mathcal{N}(\mu_b, \sigma_b^2), \quad \sigma_j \sim \text{HalfNormal}(\tau_\sigma).$$

**Relationship to the Binary Rasch Model**. The continuous IRT model can be viewed as a natural extension of the classical Rasch and 2PL models. This continuous formulation assumes that the log-odds of the observed score follow a normal distribution centered at $a_j\theta_i - b_j$, with $\sigma_j$ capturing item-specific scoring noise. As $\sigma_j \to 0$, the model collapses to the deterministic form $y_{ij} = \sigma(a_j\theta_i - b_j)$, matching the expected probability in the 2PL model. Especially when $a_j$ is fixed at 1, the correctness probability reduces to that of the Rasch model.

## 4.2 STRUCTURAL KNOWLEDGE INJECTION THROUGH FACTORIZATION TECHNIQUES

While LEGO-CM effectively broadened the applicability of IRT-based solutions. It still operates under a *single metric, single benchmark* setup, which is a simplified situation of real-world LLM capability assessment where models are tested over a wide range of benchmarks under (potentially) distinct metrics. We identify the extra-complexity brought by such more complicated evaluation scenarios as *structurally more informative*, with the following canonical formulation:

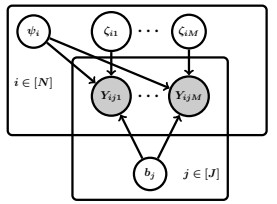

Figure 2: PGM description of LEGO-MM

**Multiple metrics** In text summarization or translation tasks, metrics like BLEU, ROUGE, and BERTScore are often used in parallel to assess the performance of language models. However, for the same item, the difficulty, discrimination, and noise level can vary across different metrics.

**Multiple benchmarks** The ability assessment of LLMs typically requires testing over several benchmarks (Qwen et al., 2025; Comanici et al., 2025) and aggregating over benchmark-specific performances using methods like averaging or win-rate (Zheng et al., 2023). However, the a priori selection of testbed benchmarks may exhibit a bias toward certain types of capabilities, rendering model evaluation results less robust. For example, for strong reasoning models that got extensively trained on mathematics and coding problems (Abdin et al., 2025; Xiaomi, 2025), their performance

over reasoning-oriented benchmarks like AIME AIME (2024) or LiveCodeBench (Jain et al., 2024) might be impressive, but their accuracy over commonsense questions might even deteriorate compared to their non-reasoning counterparts. Therefore, a more principled evaluation should efficiently decouple the general, or inherent capability of an LLM while accounting for its specificity over certain types of tasks.

To address the aforementioned challenges, we draw insights from recent developments in IRT theory (Fang et al., 2021; Chen et al., 2025) by leveraging the factorization technique. Intuitively, we factorize a model's ability into an additive combination of *general ability* and *metric-or-benchmark-specific offset*. Specifically, for the multi-metrics setup, we assume there are $N$ LLMs, $J$ items, and $M$ evaluation metrics, which we assume to take values in $[0, 1]$. The following construction, which we called LEGO-MM (MM as an abbreviation for **M**ultiple **M**etrics), effectively handles heterogeneity among distinct metrics:

$$\theta_{im} = \psi_i + \zeta_{im}$$
$$\text{logit}(Y_{ijm}) \sim \mathcal{N}(a_j\theta_{im} - b_j, \sigma_j^2). \tag{LEGO-MM}$$

Here $\theta_{im}$ is decomposed into two components, the general ability parameter $\theta_i$ and metric-specific offset parameter $\zeta_{im}$. A PGM-style description of LEGO-MM is presented in figure 2. The design is closely related to the bifactor model (Fang et al., 2021) in psychometrics, where the primary-level parameter $\psi_i$ captures the overall ability of LLM $i$, and the secondary-level parameter $\zeta_{im}$ captures the deviation in metric $m$. The parameter estimation for LEGO-MM is conducted using MCMC with the following set of priors:

$$\psi_i \sim \mathcal{N}(0, 1), \quad \boldsymbol{\zeta}_i \sim \mathcal{N}(\mathbf{0}, \Sigma_\zeta), \tag{5}$$
$$a_j \sim \text{LogNormal}(\mu_a, \sigma_a^2), \quad b_j \sim \mathcal{N}(\mu_b, \sigma_b^2), \quad \sigma_j \sim \text{HalfNormal}(\tau_\sigma),$$

where $\boldsymbol{\zeta}_i = (\zeta_{i1}, \dots, \zeta_{iM})^\top \sim \mathcal{N}(\mathbf{0}, \Sigma_\zeta)$, and $\Sigma_\zeta$ is parameterized via an LKJ-Cholesky prior (Lewandowski et al., 2009) to learn inter-metric correlations. Due to the existence of prior $\mathcal{N}(\mathbf{0}, \Sigma_\zeta)$, the estimator $\hat{\zeta}_{im}$ would be pushed towards zero, making the results more interpretable in ad-hoc analysis and more robust in prediction tasks.

Next, we propose a solution to the multi-benchmark challenge, which we call LEGO-MB (MB as an abbreviation for **M**ultiple **B**enchmarks). The design principle closely mirrors that of LEGO-MM which the primary difference being that in LEGO-MM, we factorize the LLM's latent ability along the metric dimension, while in LEGO-MB, we factorize along the benchmark dimension. Concretely, with slight overloading of notations, we assume there are $N$ LLMs and $M$ benchmarks, where benchmark $m$ contains $J_m$ items with the total number of items $J = \sum_{m=1}^{M} J_m$. Let $Y_{ij}$ denote the metric of LLM $i$ evaluated at item $j$ and $m(j)$ indicate the benchmark that $j$ belongs to. We further assume that, for different items, the response can be either continuous or binary. Let $\mathcal{J}_c$ denote the set of continuous-score items and $\mathcal{J}_b$ the set of binary items, with $\mathcal{J}_c \cup \mathcal{J}_b = \{1, \dots, J\}$ and $\mathcal{J}_c \cap \mathcal{J}_b = \emptyset$. The following set of equations characterizes the construction of LEGO-MB:

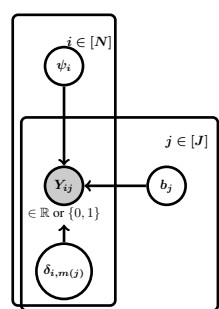

Figure 3: PGM description of LEGO-MB

$$\theta_{i,m} = \psi_i + \delta_{i,m}, \quad m = 1, \dots, M.$$
$$\begin{cases} \text{logit}(Y_{ij}) \sim \mathcal{N}(a_j\theta_{i,m(j)} - b_j, \ \sigma_j^2), & \text{if } j \in \mathcal{J}_c \\ Y_{ij} \sim \text{Bernoulli}(\sigma(a_j\theta_{i,m(j)} - b_j)), & \text{if } j \in \mathcal{J}_b \end{cases}. \tag{LEGO-MB}$$

The proposed LEGO-MB model can simultaneously handle multiple benchmarks with different sizes and items with different metric types (i.e., binary or continuous). Note that the latent ability is both LLM and benchmark-dependent. This model with fine-grained structure is again an extended version of the multi-dimensional IRT model. The covariance matrix $\Sigma_\delta$ here explicitly characterizes relationships across benchmarks. Positive correlations indicate that two benchmarks tend to assess similar abilities of LLMs, whereas negative correlations reveal intrinsic differences between the design of benchmarks. We hope that such kinds of principled methods may shed light on optimized designs of benchmarks. The MCMC estimation of LEGO-MB is similar to that of LEGO-MM, with

the following priors,

$$\psi_i \sim \mathcal{N}(0,1), \quad \boldsymbol{\delta}_i \sim \mathcal{N}(\mathbf{0}, \Sigma_\delta), \tag{6}$$

$$a_j \sim \text{LogNormal}(\mu_a, \sigma_a^2), \quad b_j \sim \mathcal{N}(\mu_b, \sigma_b^2), \quad \sigma_j \sim \text{HalfNormal}(\tau_\sigma),$$

where $\boldsymbol{\delta}_i = (\delta_{i1}, \ldots, \delta_{iM})^\top \sim \mathcal{N}(\mathbf{0}, \Sigma_\delta)$, and $\Sigma_\delta$ is also parameterized via an LKJ-Cholesky prior to learn inter-benchmark correlations.

## 5 EXPERIMENTS

In this section, we present empirical investigations showing the effectiveness of the proposed LEGO-IRT framework, focusing on three aspects:

- **Accuracy of ability estimation**: We follow approaches from previous work to use LLM performance prediction for measuring ability estimation accuracy in LEGO-IRT. Higher predictive accuracy shows the IRT model captures LLM ability more precisley.
- **Stability and data efficiency**: We test LEGO-IRT's stability using randomly sampled subsets and examine the smallest portion of items needed for stable scoring, measuring its data efficiency.
- **Structural benefits**: We investigate more complex scenarios where several distinct metrics are applied to the same benchmark, as well as scenarios where multiple benchmarks could be modeled simultaneously for performance improvements.

### 5.1 EXPERIMENTAL SETUP

We use three comprehension-type benchmarks comprising multiple choice questions (MCQ): MMLU (Hendrycks et al., 2021), CSQA (Talmor et al., 2019), and Ceval (Huang et al., 2023), among which MMLU and CSQA are English benchmarks while Ceval focuses on answering questions using Chinese. We also use two seq2seq-type benchmarks: XSUM (Narayan et al., 2018), which evaluates text summarization abilities, and WMT20 (Mathur et al., 2020), where we use the [cs→en] subtask that inspects Chinese-to-English translation capability. For two seq2seq benchmarks, we consider 7 metrics which all take values in the range $[0, 1]$: ROUGE-style metrics (Lin, 2004) $\text{ROUGE}_1$, $\text{ROUGE}_L$, METEOR (Szpektor et al., 2007), BLEU (Papineni et al., 2002), and three BERTScore family metrics $\text{BERTScore}_P$, $\text{BERTScore}_R$ and $\text{BERTScore}_{F1}$ (Zhang et al., 2019). As we operate on a tight budget, for all the 5 involved benchmarks, we select subsets of a reasonable size to allow for an affordable inference cost. A detailed summary of the benchmarks is listed in Table 4 in Appendix B.1, with a total of 8,918 question items.

We evaluate 70 models (The complete list is reported in table 5) over the selected benchmarks, ranging from the latest frontier LLMs, which have strong reasoning capabilities (OpenAI, 2025; Comanici et al., 2025; Anthropic, 2025) to state-of-the-art open weight model series that exhibit a fine-grained hierarchy of capability scaling (Qwen et al., 2025; Yang et al., 2025). The environments we used for inference are detailed in appendix B.1. We have found the chosen set of models to cover a broad spectrum of model capabilities, constituting a suitable pool of test takers.

### 5.2 ACCURACY AND STABILITY ASSESSMENTS OVER SEQ2SEQ TASKS

In this section, we investigate LEGO-IRT over two benchmarks with seq2seq style: XSUM and WMT20.

**Accuracy assessments** To inspect performance prediction accuracy, we first select $10\%$ of all the problem-specific individual model metrics as a held-out test set, and randomly select $r$ proportion of the remaining as our training set, where we allow $r$ to vary $r \in \{0.1, \ldots, 1.0\}$. 3 independent random trials are applied to each training configuration.

We compare LEGO-IRT (using the implementation of LEGO-CM) with three alternative baselines over the test set: global mean estimation (GME) where we use the average of the entire training set as the ad-hoc prediction; model mean estimation (MME) where we perform a stratified estimation divided by specific models and item mean estimation (IME) where the stratification factor is the items. We use mean square error (MSE) between the predicted metrics and the oracle values as the comparison criterion under the BLEU metric. The results are shown in figure 4. The results exhibit that LEGO-IRT dominates the baselines across two benchmarks, especially in the data-abundant

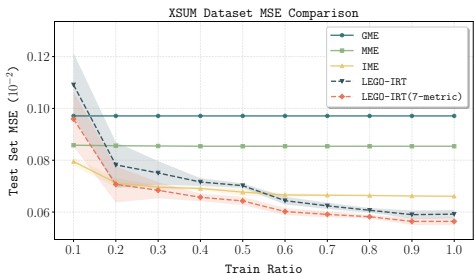 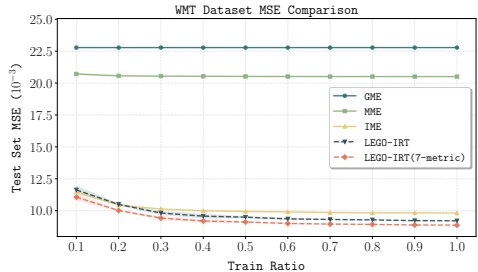

Figure 4: LLM performance prediction over XSUM and WMT20 under varying training ratios.

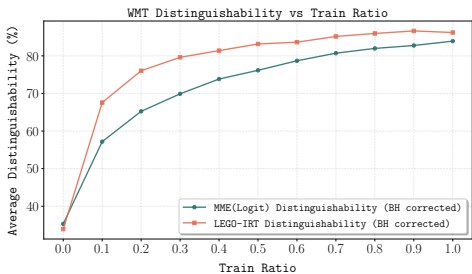 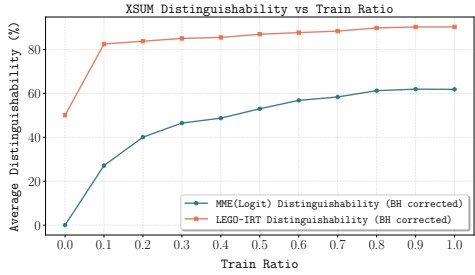

Figure 5: Power comparison between LEGO-IRT and model mean estimation (MME) over XSUM and WMT20 benchmarks.

regimes where a clear separation is evidenced. Additionally, we further demonstrate that **LEGO-IRT offers stronger statistical power for model comparisons**: Specifically, for each training run, we conduct the following (multiple) hypothesis tests across all LLM-to-LLM pairs:

$$H_0 : \theta_i = \theta_j \qquad v.s. \qquad H_1 : \theta_i < \theta_j, \qquad \forall 1 \le i, j \le M, \tag{7}$$

and use the Benjamini-Hochberg procedure (Benjamini & Hochberg, 1995) to select all statistically significant pairs while controlling the false discovery rate at $0.05$. The detailed algorithm is presented in algorithm 1 in appendix B.2. Upon obtaining the selected pairs, we filter a subset of models that admits statistically validate rankings, i.e., $i_{(1)} \prec i_{(2)} \prec \cdots \prec i_{(M^*)}$, and define *distinguishability* as the ratio $\frac{M^*}{M}$. Intuitively, distinguishability measures the proportion of models that could be compared statistically. We compare distinguishabilities at aforementioned training ratios and plot them in figure 5, where we observe a clear dominance of LEGO-IRT over model mean estimation across all training configurations, with up to $13\%$ increase in absolute scale, suggesting that LEGO-IRT offers more statistical power when applied to model comparisons.

**Stability assessments** We inspect the stability of LEGO-IRT under two perspectives: ability estimation stability and ranking stability. To test the stability of ability estimation, we randomly sampled 50 subsets from XSUM and WMT20, each containing 100 items. The resulting distribution of estimated capability for model `Gemini-2.5-pro` is plotted in figure 6a(See also figure **??** for illustration with more models.), along with estimations produced from the entire benchmark, which we referred to as global estimations. The results exhibit a much better concentration around its global estimation for LEGO-IRT than that of MME.

Next, we test the ranking stability of LEGO-IRT over the aforementioned 50 subsets using the following procedure: For any given subset, we pick five models which are conventionally perceived to be of different capability levels and treat them as *newcomer* models, with the rest being referred to as *existing* models. The intuition is to compare the ability estimates of newcomer models produced by two distinct methods: (i) through a scoring procedure after calibration using only the 65 existing models. (ii) through a joint calibration using all 70 models. The goal of this comparison is to inspect **whether item difficulties calibrated by existing models generalize to newcomers**. We use ranks of newcomer models' ability estimates among the 70 models as the base for the criterion, and compute the *ranking bias* as the absolute deviation of newcomer ranks between those produced by method (i) and (ii) as mentioned above (regarded as *estimation* and *orcale*, respectively). A more

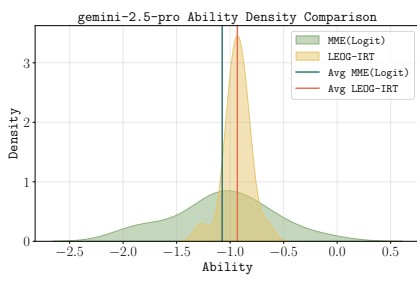

(a) Stability of ability estimation across random subsets.

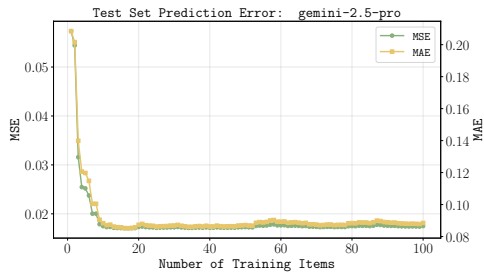

(b) Stability of performance prediction under varying number of scoring items.

Figure 6: Stability assessment of `Gemini-2.5-pro` on XSUM dataset.

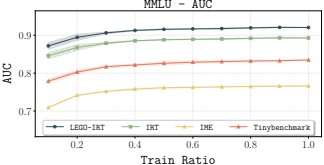
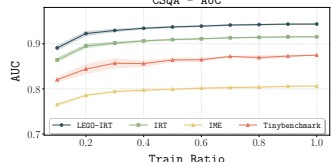
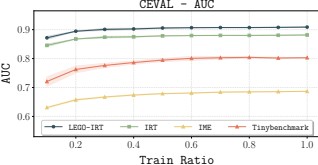

Figure 7: LLM performance prediction comparisons over MMLU, CSQA and Ceval

algorithmic description is presented in algorithm 2 in appendix B.2. The results are presented in table 2, showing that LEGO-IRT exhibits a smaller ranking bias overall, as well as achieving better stability as showcased by smaller variations among random subsets.

**Data efficiency** To investigate the data efficiency during *scoring stage*, we use the same set of newcomer models as in the previous section. The item difficulties are fixed at their calibrated levels using the remaining 65 models' response metrics. Then we gradually increase the number of items for scoring the newcomer models' capabilities, similar to that in (Truong et al., 2025, Section 3). The estimated model abilities are verified on a hold-out test set using either the MSE or MAE metric. We report the result in figure 6b. The result suggests that **utilizing** 50 **items—accounting for merely 3% of the total item pool**—the test set prediction error stabilizes and the ability estimates converge. **This demonstrates that LEGO-IRT, calibrated on an existing item bank, can reliably project new models onto a unified ability scale even under extremely sparse response observations.**

Table 2: Absolute rank deviations and their standard deviation between MME and LEGO-IRT.

| Model | XSUM | | WMT20 | |
|---|---|---|---|---|
| | MME | LEGO-IRT | MME | LEGO-IRT |
| Gemini-2.5-pro | $8.40_{\pm 17.58}$ | $\mathbf{1.20}_{\pm 2.69}$ | $0.85_{\pm 4.51}$ | $\mathbf{0.80}_{\pm 4.08}$ |
| GPT-4.1-2025-04-14 | $7.90_{\pm 17.80}$ | $\mathbf{0.10}_{\pm 6.98}$ | $1.10_{\pm 12.44}$ | $\mathbf{0.20}_{\pm 10.35}$ |
| Qwen3-8B | $7.60_{\pm 17.36}$ | $\mathbf{1.60}_{\pm 9.58}$ | $3.70_{\pm 10.41}$ | $\mathbf{3.15}_{\pm 7.37}$ |
| Qwen3-1.7B | $11.80_{\pm 20.79}$ | $\mathbf{2.65}_{\pm 7.03}$ | $1.55_{\pm 4.14}$ | $\mathbf{0.55}_{\pm 2.81}$ |
| Qwen2.5-0.5B-Instruct | $\mathbf{1.95}_{\pm 14.74}$ | $5.40_{\pm 8.79}$ | $\mathbf{0.90}_{\pm 10.00}$ | $1.80_{\pm 5.90}$ |

## 5.3 EXPLORATION OF STRUCTURAL BENEFITS

In this section, we inspect the advantage of incorporating additional structural information through the lens of LLM performance prediction.

**Benefits of joint modeling between distinct metrics** We investigate the LEGO-MM model developed in section 4.2 in an analogous model setup in the previous section, where we assess predictive performance under varying training data ratios. In addition to the single-metric LEGO-CM model, we fit a multi-metric LEGO-MM model that integrates 7 metrics (presented as LEGO-IRT (7 met-

|  | MEAN$_{\text{MMLU}}$ | MEAN$_{\text{CSQA}}$ | MEAN$_{\text{Ceval}}$ | $\psi_{\text{LEGO-IRT}}$ |
|---|---|---|---|---|
| Spearman's $\rho$ | 0.809 | 0.615 | 0.812 | **0.836** |
| Kendall's $\tau$ | 0.633 | 0.447 | 0.641 | **0.651** |

Table 3: Rank correlation measures between estimated ability and the `Text` section score in LMArena (Chiang et al., 2024). For example, MEAN$_{\text{MMLU}}$ stands for the correlation between test models' average scores on MMLU and their associating LMArena score.

rics) in figure 4). The results are presented in figure 4, demonstrating that integrating multiple metrics further enhances the predictive performance of LEGO-CM, suggesting that the correlation structures between distinct metrics are effectively exploited. Meanwhile, as the model LEGO-MM also allows explicit modeling and interpretation of metric correlations, we conduct a detailed correlation analysis between all the 7 metrics, which is postponed to appendix C.2. An interesting finding therein is that after adjusting for confounding factors such as problem difficulty and generic model ability, there exists a negative correlation (measured in terms of $\zeta$ as in the definition of equation LEGO-MM) between ngram-based metrics like ROUGE and semantic-inspired metrics like BERTScore.

**Benefits of integrating multiple benchmarks** We investigate the LEGO-MB model developed in section 4.2 to two multi-benchmark scenarios. In the first experiment, we use three benchmarks under binary metric: MMLU, CSQA, and Ceval. We adopt a similar experiment methodology as in section 5.2 by varying training data ratios from $r \in \{0.2, 0.3, \ldots, 1.0\}$. Under each training configuration, we use all the training samples from the three benchmarks to train our LEGO-MB model that exploits the inter-benchmark correlations. We additionally compare with two baselines: The method used in the tinyBenchmarks paper (Polo et al., 2024), and the standard IRT model used in (Truong et al., 2025). We use AUC as the criterion for performance prediction. The results are depicted in figure 7. The results suggest a solid improvement of LEGO-IRT over standard IRT approaches, illustrating the advantage of incorporating inter-benchmark correlations. Additionally, we present our findings regarding the detailed correlation structure between the three benchmarks in appendix C.3, demonstrating the strong dependence among English comprehension tasks and a much weaker correlation between English comprehension and Chinese comprehension tasks. Additionally, we postpone a report of integrating continuous benchmarks to appendix C.4, where analogous benefits are observed.

Finally, we explore potential interpretations of the estimated general capability $\psi$ as defined in equation LEGO-MB. As there are no gold standards for model ability, we use the human-judged, widely accepted authentic score from the LMArena(`Text`) leaderboard Chiang et al. (2024) as a reasonable surrogate. [2] Among the 70 test models, 40 of them have voting results in LMArena. We compare all those corresponding estimated $\psi$ capabilities with mean score aggregation over MMLU, CSQA, and Ceval, measured by rank correlation with LMArena score, which are detailed in table 3. The results reveal an interesting finding that by leveraging principled evaluation paradigms like LEGO-IRT, we obtain (latent) model ability characterizations that might exhibit better alignment with human judgments.

## 6 CONCLUSION

We introduced LEGO-IRT, a unified framework for data-efficient LLM evaluation that overcomes the limitations of prior item response theory (IRT) approaches. Our framework uniquely supports both binary and continuous metrics natively, eliminating the need for information-losing discretization. Through a novel factorized design, LEGO-IRT incorporates structural knowledge by jointly modeling multiple metrics and benchmarks. Our extensive experiments show that LEGO-IRT provides stable capability estimates using as little as $3\%$ of the full evaluation data. Furthermore, leveraging structural information reduces estimation error by up to $10\%$. This work marks a significant step towards more affordable, reliable, and nuanced LLM assessment.

---

[2]`https://lmarena.ai/leaderboard/text`. We pick the data snapshot as of September 18, 2025.

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

# Appendices of "Toward a Unified Framework for Data- Efficient Evaluation of Large Language Models"

## A DISCLOSURE: USE OF LLMS

LLMs are used as an assistant in this project. Specifically, LLMs are used for

- Polish writing: Refine statements, improve fluency.
- Coding assistance: Help draft prototype codes and alleviate engineering efforts.

## B EXPERIMENTAL DETAILS

### B.1 BENCHMARKS AND MODELS

A summary for all five datasets we used is reported in table 4

| Benchmark | MMLU | CSQA | Ceval | XSUM | WMT20 |
|-----------|------|------|-------|------|-------|
| Question type | MCQ | MCQ | MCQ | NLG | NLG |
| # items | 2035 | 1221 | 2194 | 2050 | 1418 |

Table 4: Summary statistics for the evaluation benchmarks. MCQ denotes **M**ultiple **C**hoice **Q**uestion, NLG denotes **N**atural **L**anguage **G**eneration

We report all 70 models used in our experiments in table 5. While some of the models support *thinking* mode, we by default disable the thinking process if it is configurable for the model to save inferential cost. For open models of small size, i.e., model size smaller than $20B$, we use two NVidia RTX 3090 GPUs to conduct inference using the vLLM Kwon et al. (2023) framework. The inference time configurations follow the default open-source recipe if available. For models of greater size or proprietary models, we call official APIs with a final total api cost of $887.

Table 5: The complete list of models used in empirical investigations.

| Model Name | Model Size (B) | Proprietary or Open | Requires Thinking |
|-----------|----------------|---------------------|-------------------|
| Hunyuan-0.5B-Instruct | 0.5 | Open | Yes |
| Hunyuan-1.8B-Instruct | 1.8 | Open | Yes |
| Hunyuan-4B-Instruct | 4 | Open | Yes |
| Hunyuan-7B-Instruct | 7 | Open | Yes |
| MiMo-7B-SFT | 7 | Open | Yes |
| Minimax-m1 | 456 | Open | No |
| MiniMax-Text-01 | Unknown | Proprietary | No |
| Mistral-7B-Instruct-v0.3 | 7 | Open | No |
| Mixtral-8x7B-Instruct-v0.1 | 47 | Open | No |
| NVIDIA-Nemotron-Nano-12B-v2 | 12 | Open | Yes |
| NVIDIA-Nemotron-Nano-9B-v2 | 9 | Open | Yes |
| Qwen2.5-0.5B-Instruct | 0.5 | Open | No |
| Qwen2.5-1.5B-Instruct | 1.5 | Open | No |
| Qwen2.5-3B-Instruct | 3 | Open | No |
| Qwen2.5-7B-Instruct | 7 | Open | No |
| Qwen2.5-14B-Instruct | 14 | Open | No |
| Qwen2.5-72B-Instruct | 72 | Open | No |
| Qwen3-0.6B | 0.6 | Open | Yes |
| Qwen3-1.7B | 1.7 | Open | Yes |
| Qwen3-4B | 4 | Open | Yes |

**Continued from previous page**

| Model Name | Model Size (B) | Proprietary or Open | Supports Thinking |
|---|---|---|---|
| `Qwen3-8B` | 8 | Open | Yes |
| `Qwen3-14B` | 14 | Open | Yes |
| `Qwen3-32b` | 32 | Open | Yes |
| `Qwen3-235b-a22b-thinking-2507` | 235 | Open | Yes |
| `Qwen-max-2025-01-25` | Unknown | Proprietary | No |
| `Qwen-plus` | Unknown | Proprietary | No |
| `Qwen-turbo` | Unknown | Proprietary | No |
| `Claude-opus-4` | Unknown | Proprietary | Yes |
| `Claude-sonnet-4` | Unknown | Proprietary | Yes |
| `Claude-3-5-haiku-20241022` | Unknown | Proprietary | No |
| `Claude-3-5-sonnet-20241022` | Unknown | Proprietary | No |
| `Claude-3-haiku-20240307` | Unknown | Proprietary | No |
| `Claude-3-opus-20240229` | Unknown | Proprietary | No |
| `Deepseek-chat-v3.1` | 660 | Open | Yes |
| `Gemma-2-2b-it` | 2 | Open | No |
| `Gemma-2-9b-it` | 9 | Open | No |
| `Gemma-3-12b-it` | 12 | Open | No |
| `Gemma-3-1b-it` | 1 | Open | No |
| `Gemma-3-4b-it` | 4 | Open | No |
| `Gemma-3-27b-it` | 27 | Open | No |
| `Gemini-2.0-flash` | Unknown | Proprietary | No |
| `Gemini-2.5-flash` | Unknown | Proprietary | Yes |
| `Gemini-2.5-pro` | Unknown | Proprietary | Yes |
| `GPT-4o-2024-05-13` | Unknown | Proprietary | Yes |
| `GPT-4o-mini-2024-07-18` | Unknown | Proprietary | Yes |
| `GPT-oss-20B` | 20 | Open | Yes |
| `GPT-oss-120b` | 120 | Open | Yes |
| `GPT-4.1-2025-04-14` | Unknown | Proprietary | Yes |
| `GPT-4.1-mini-2025-04-14` | Unknown | Proprietary | Yes |
| `GPT-5-2025-08-07` | Unknown | Proprietary | Yes |
| `GPT-5-mini-2025-08-07` | Unknown | Proprietary | Yes |
| `GPT-5-nano-2025-08-07` | Unknown | Proprietary | Yes |
| `Internlm3-8b-instruct` | 8 | Open | No |
| `Llama-3.1-8B-Instruct` | 8 | Open | No |
| `Llama-3.3-70B-Instruct` | 70 | Open | No |
| `Llama-3.1-405B-Instruct` | 405 | Open | No |
| `Llama-3.1-70B-Instruct` | 70 | Open | No |
| `Llama-4-maverick` | Unknown | Open | No |
| `Llama-4-scout` | Unknown | Open | No |
| `Kimi-k2-preview` | Unknown | Proprietary | No |
| `Phi-3.5-mini-instruct` | 3.8 | Open | No |
| `Phi-4` | 14 | Open | No |
| `Phi-4-mini-instruct` | 3.8 | Open | No |
| `Phi-4-reasoning-plus` | 14 | Open | Yes |
| `Grok-3-beta` | Unknown | Proprietary | No |
| `Grok-3-mini-beta` | Unknown | Proprietary | No |
| `Grok-4-07-09` | Unknown | Proprietary | Yes |
| `GLM-4-9B-0414` | 9 | Open | No |
| `GLM-4.5` | 355 | Open | Yes |
| `GLM-4.5-air` | 106 | Open | Yes |

## B.2 ALGORITHM DESCRIPTIONS

In this section, we provide the missing details of the main context.

Algorithm 1 summarizes the complete procedure of computing the model ability discriminability, where we use the $t$-test (naive method) or the $z$-test (our LEGO-IRT) to compute the p-values for each pair of models and identify those distinguishable pairs via the Benjamini-Hochberg method. We report the average number of detected pairs as the final score. A larger score indicates better performance.

---

**Algorithm 1** Model Ability Discriminability Evaluation

---

**Input:** A collection of models $\{M_1, M_2, \ldots, M_N\}$, two ability estimation procedures: Method 1 (logit mean response approach) and Method 2 (LEGO-IRT estimation), and a significance threshold $\alpha \in (0, 1)$.

**Output:** Discriminability scores $R^{(1)}$ and $R^{(2)}$ quantifying the ability of each method to distinguish models.

1: **for** $i = 1$ **to** $N$ **do**
2:     For each competing model $M_j$, $j \neq i$, compute the pairwise $p$-values under both methods:
3:         **Method 1**: Apply logit transformation to response values, estimate means and variances, then perform two-sample $t$-tests:

$$p_{i,j}^{(1)} = \text{t-test}\big(\text{logit}(M_i), \text{logit}(M_j)\big)$$

4:         **Method 2**: Estimate ability parameters $\hat{\theta}$ and standard errors via LEGO-IRT; compute standardized $z$-scores and corresponding two-sided $p$-values:

$$z_{i,j} = \frac{\hat{\theta}_i - \hat{\theta}_j}{\sqrt{\text{SE}_i^2 + \text{SE}_j^2}}, \quad p_{i,j}^{(2)} = 2\Phi(-|z_{i,j}|)$$

5:         (Item difficulty parameters are fixed based on response calibration from other models.)
6:     Apply the Benjamini-Hochberg (BH) procedure to control the false discovery rate (FDR) for each set of $p$-values $\{p_{i,j}^{(1)}\}_{j \neq i}$ and $\{p_{i,j}^{(2)}\}_{j \neq i}$:
7:         Sort the $p$-values in ascending order: $p_{(1)} \leq p_{(2)} \leq \cdots \leq p_{(m)}$, where $m = N - 1$.
8:         Find the maximal index $k$ satisfying

$$p_{(k)} \leq \frac{k}{m}\alpha$$

9:         Reject all null hypotheses corresponding to $p_{(i)}$ for $i \leq k$, indicating significant ability differences.
10:     Compute the number of significantly distinguishable models $S_i^{(1)}$ and $S_i^{(2)}$ post-correction.
11:     Calculate the proportion of distinguishable models for each method:

$$r_i^{(1)} = \frac{S_i^{(1)}}{N-1}, \quad r_i^{(2)} = \frac{S_i^{(2)}}{N-1}$$

12: **end for**
13: Aggregate the overall discriminability scores by averaging across all models:

$$R^{(1)} = \frac{1}{N}\sum_{i=1}^{N} r_i^{(1)}, \quad R^{(2)} = \frac{1}{N}\sum_{i=1}^{N} r_i^{(2)}$$

14: **return** Discriminability metrics $R^{(1)}$ and $R^{(2)}$.

---

Algorithm 2 summarizes the detailed procedure of evaluating the new model, where we use the full responses to estimate the true model ability and bootstrap subsets of responses to examine the robustness of the estimation ability of the naive method and the proposed LEGO-IRT. We report the mean and standard deviation of the rank difference. Smaller values indicate better performances.

---

**Algorithm 2** Ability Estimation and Rank Deviation Analysis for a Novel Model

---

**Input:** Existing model set with response data and combined ability estimates, response data of the novel model, subset size $M$, number of sampling iterations $K$

**Output:** Mean and standard deviation of rank deviations for the novel model under both estimation methods.

1: Compute the ground truth rank $r^*$ of the novel model by:
2:    Estimating ability using full response data via Method 1 (logit mean response) and Method 2 (LEGO-IRT).
3:    Combining the two ranks to define $r^*$.
4: **for** $k = 1$ **to** $K$ **do**
5:    Randomly sample a subset of items of size $M$ from the full item pool.
6:    Estimate the novel model's ability on the subset using Method 1 and Method 2, yielding $\hat{\theta}_k^{(1)}$ and $\hat{\theta}_k^{(2)}$.
7:    Determine the rank of the novel model among all models based on $\hat{\theta}_k^{(1)}$ and $\hat{\theta}_k^{(2)}$, denoted $r_k^{(1)}$ and $r_k^{(2)}$.
8: **end for**
9: Calculate rank deviations for each method:
$$d_k^{(m)} = r_k^{(m)} - r^*, \quad m = 1, 2$$

10: Compute the mean and standard deviation of rank deviations:
$$\bar{d}^{(m)} = \frac{1}{K} \sum_{k=1}^{K} d_k^{(m)}, \quad \sigma^{(m)} = \sqrt{\frac{1}{K-1} \sum_{k=1}^{K} \left(d_k^{(m)} - \bar{d}^{(m)}\right)^2}, \quad m = 1, 2$$

11: **return** $\bar{d}^{(1)}, \sigma^{(1)}, \bar{d}^{(2)}, \sigma^{(2)}$.

---

# C ADDITIONAL EXPERIMENTAL RESULTS

In this section, we present additional experimental results that further validate the effectiveness, stability, and interpretability of the LEGO-CM, LEGO-MM, and LEGO-MB models.

## C.1 STATISTICAL STRENGTH OF LEGO-CM

We first examine the statistical power of LEGO-CM in model comparisons. While Figure 5 in the main text illustrates distinguishability after BH correction, Figure 8 supplements this by showing the uncorrected distinguishability performance across varying training ratios. This comparison highlights the robustness of LEGO-CM 's testing power whenever the multiple testing correction procedure is applied or not.

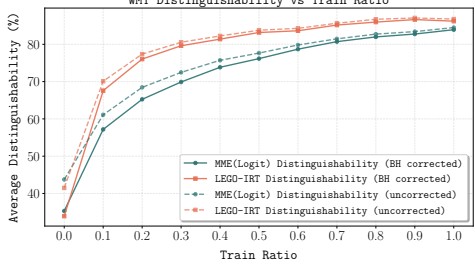 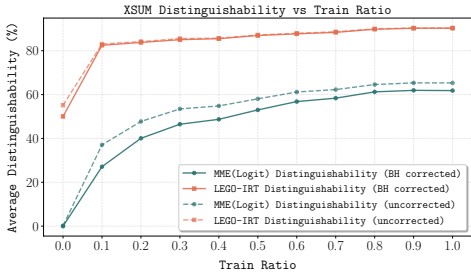

Figure 8: Power comparison between LEGO-IRT and model mean estimation (MME) over XSUM and WMT20 benchmarks, shown before and after BH correction. LEGO-IRT always achieves better results.

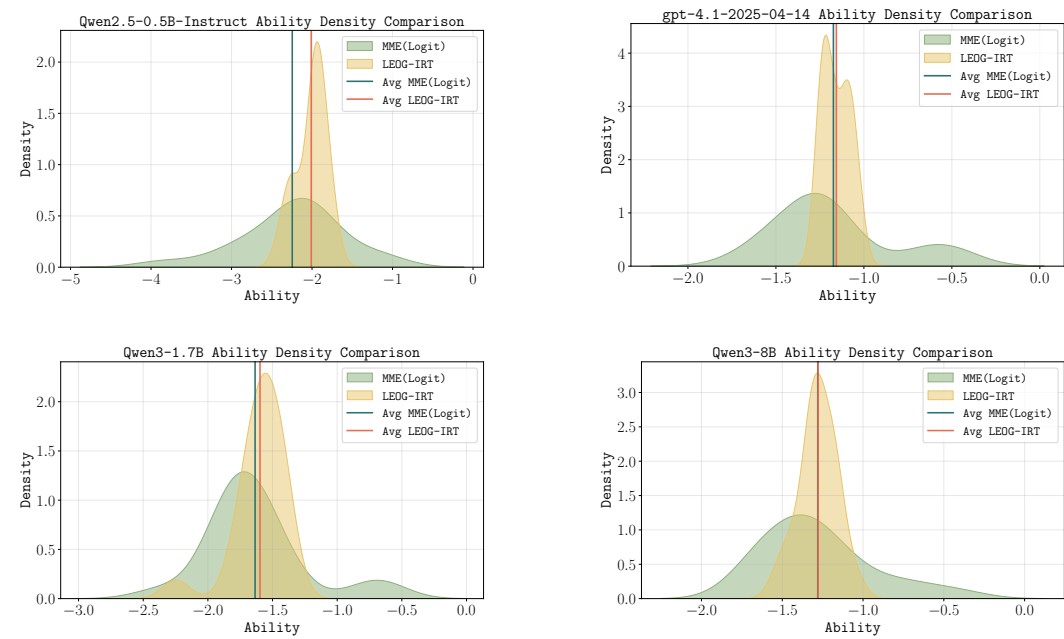

Figure 9: Stability assessment of LLM models on XSUM dataset.

In addition to the stability analysis of the Gemini-2.5-pro shown in the main text (Figure 6), we extend this evaluation to four additional models. Figures 9 and 10 show the density of the estimated LLM's latent ability and convergence behaviors, respectively. These results demonstrate that LEGO-CM consistently achieves robust estimates and reliable model rankings across diverse models, underscoring its general applicability.

Figure 11 shows that, as the training ratio increases, 95% credible intervals of latent ability of five representative LLMs become non-overlapping. This phenomenon confirms that MCMC successfully provides the statistically significant results, validating that the uncertainty of latent ability estimation can be effectively reduced within the LEGO-CM framework. According to the figure, the abilities of five models can be reliably ranked with statistically significant differences using only 40% of the training data, which cannot be achieved by using the aggregated mean alone.

## C.2 STRONG INTERPRETATION OF LEGO-MM

We further analyze the LEGO-MM 's ability estimates and metric correlation patterns to deepen understanding of multi-metric evaluation.

To provide readers with a clearer global picture, Figure 12 displays a heatmap of 70 LLM abilities across seven metrics, with color intensity reflecting ability magnitude. This visualization facilitates direct comparison of model strengths on each metric. To be more detailed, Figure 13a shows radar charts of three representative models, illustrating their overall and metric-specific abilities. Notably, qwen-turbo and qwen3-235b-a22b-thinking-2507 demonstrate consistent performance across metrics, while qwen3-32b1.8 excels particularly on BERTScore-R and METEOR.

To quantify sensitivity, we compute the standard deviation of metric-specific parameter $\zeta_{im}$'s for LLM $i$ and rank the values from the lowest to the highest. Claude-opus-4 and Gemini-2.5-pro exhibit the lowest sensitivities (1.98 and 1.97), ranking first and fourth in global ability (i.e., $\psi_i$), respectively. This highlights their robust and stable performance in text generation tasks. To be self-complete, we also provide Figure 13b to show the density of standard deviations of $\zeta_{im}$'s of all 70 LLMs.

Figure 14 contains two correlation matrices that illustrate the metric interrelations estimated under the LEGO-MM framework. Unlike traditional post-hoc Pearson correlation analysis, where metric

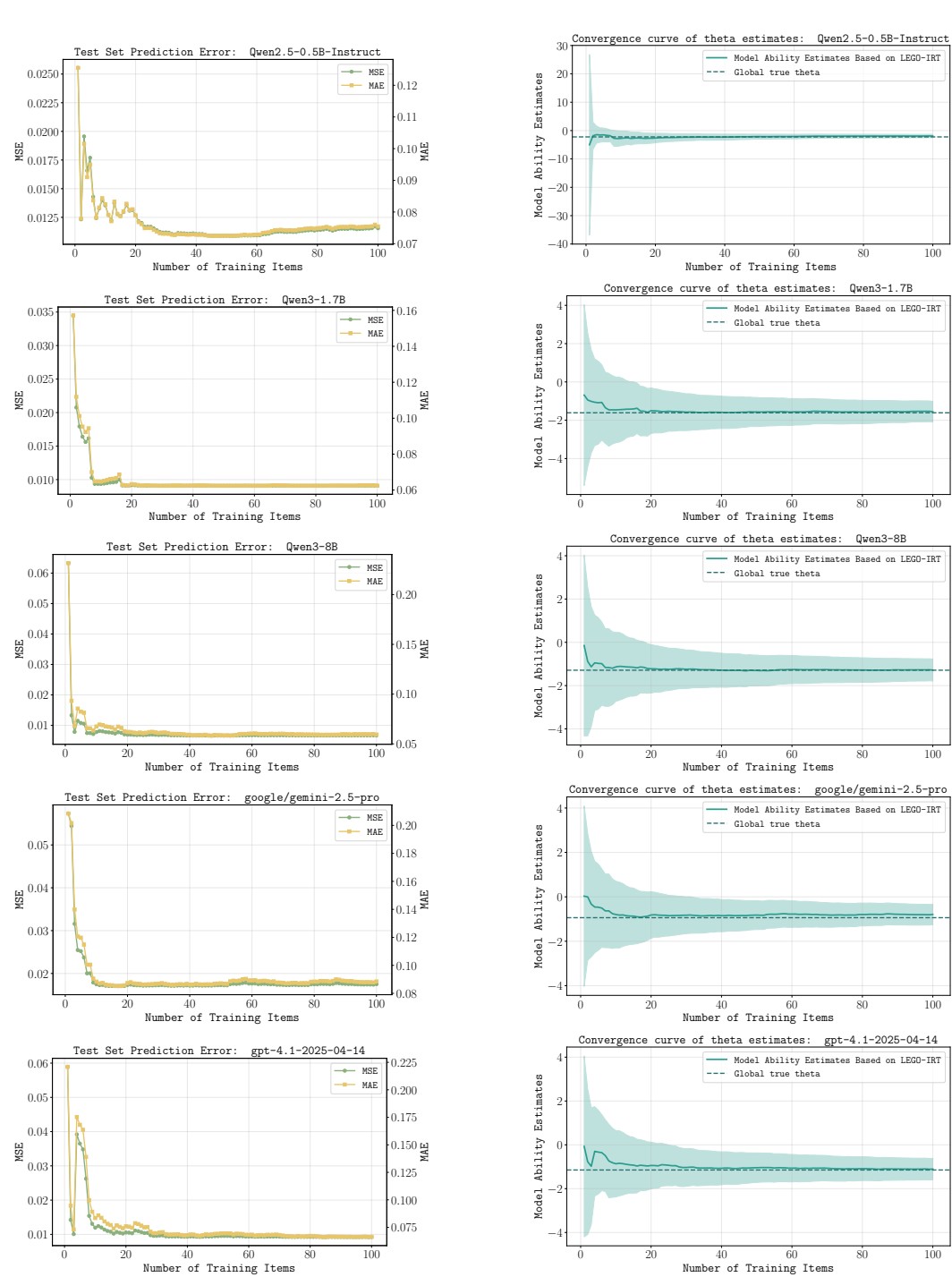

Figure 10: Convergence of estimated ability values and test set MSE for different models during ability estimation.

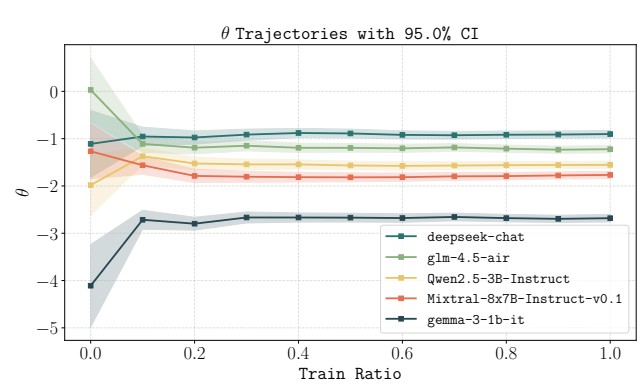

Figure 11: Ability Estimates and 95% HDI Across Training Ratios

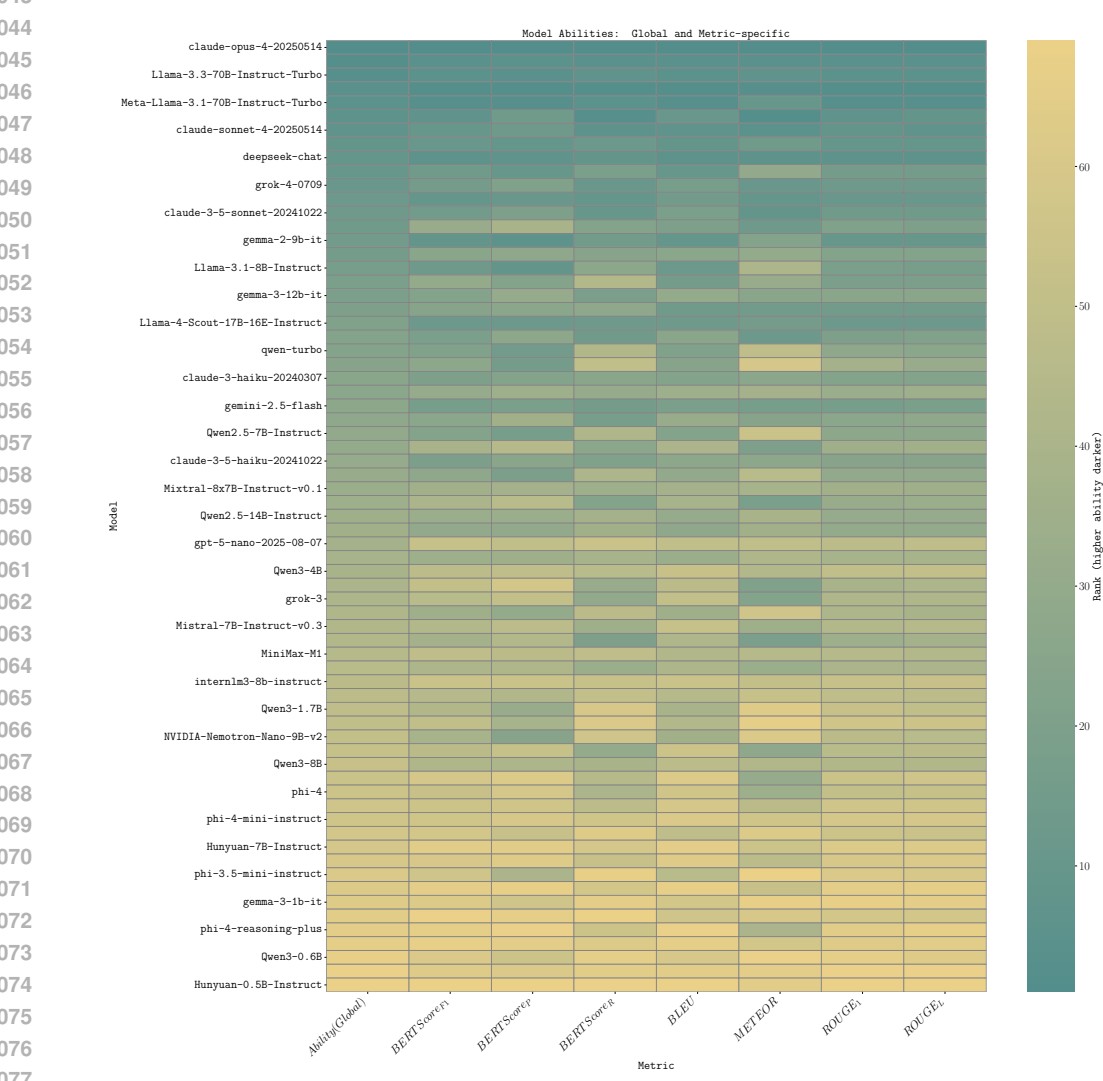

Figure 12: Heatmap of Model Ability Estimates Across Multiple Metrics

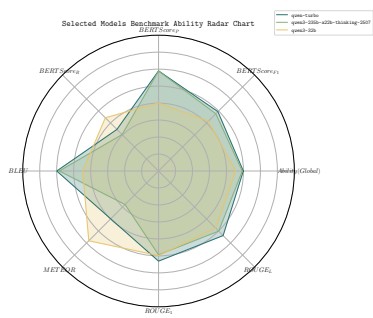
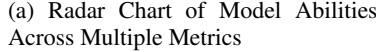

(a) Radar Chart of Model Abilities Across Multiple Metrics

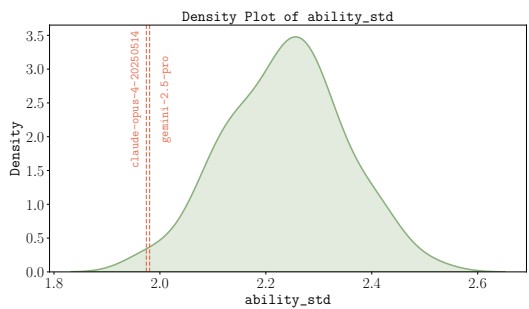

(b) Density Plot of Model Ability Stability Measured by Standard Deviation in LEGO-MM

Figure 13: Model Ability Profiles and Stability Analysis in LEGO-MM

correlations are confounded by global model ability and item difficulty effects, often resulting in uniformly positive correlations. Our approach explicitly models and removes these global factors. This yields a purified residual correlation matrix revealing more diagnostic patterns: the ROUGE family and METEOR remain strongly positively correlated, reflecting their shared focus on surface overlap and extractive consistency. In contrast, BLEU and BERTScore exhibit a negative correlation, suggesting potential antagonism between $n$-gram precision and semantic similarity dimensions.

These findings demonstrate that **LEGO-MM can effectively capture true relationships among evaluation metrics, while the naive Pearson correlation often leads to the overestimation!** This insight can guide the design of more discriminative and interpretable multi-dimensional metric systems, for example, by combining negatively correlated metrics as complementary dimensions to better capture diverse aspects of text quality.

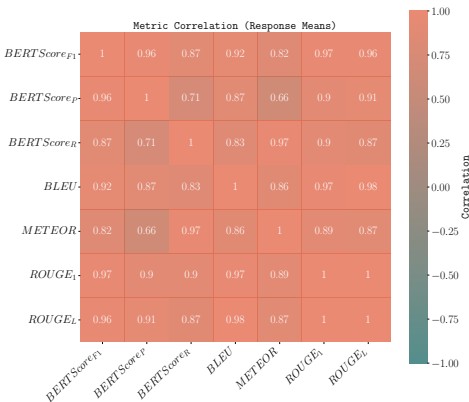
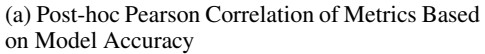

(a) Post-hoc Pearson Correlation of Metrics Based on Model Accuracy

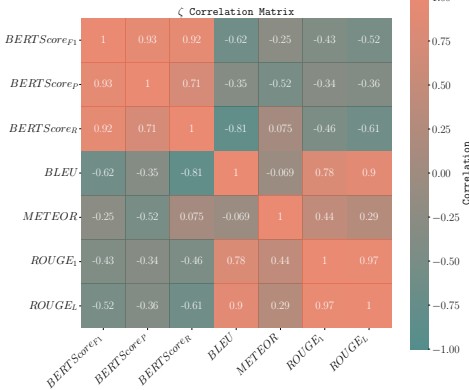

(b) Residual Correlation Matrix of Metric-specific Effects Estimated by LEGO-MM

Figure 14: Metric Interrelations: Comparison Between Post-hoc and LEGO-MM Modeled Correlations

## C.3 BETTER INSIGHTS FROM LEGO-MB

Finally, we examine the LEGO-MB model's ability estimates and inter-benchmark correlation structures to better understand multi-benchmark evaluation.

We evaluate the predictive accuracy of the LEGO-MB compared to baseline methods on three binary benchmarks: MMLU, CSQA, and Ceval. The experimental setup mirrors that of figure 7, varying training data ratios from $r \in \{0.2, 0.3, \ldots, 1.0\}$. Figure 15 shows that LEGO-MB consistently

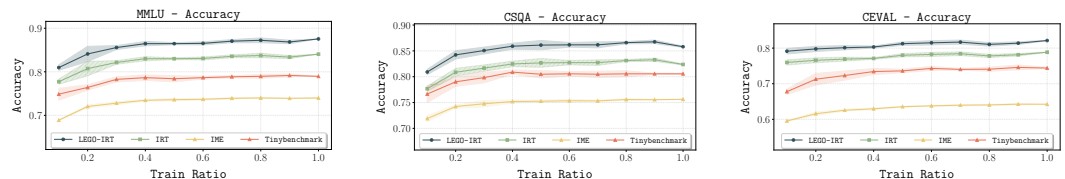

Figure 15: LLM performance prediction comparisons over MMLU, CSQA and Ceval

outperforms baselines, confirming that exploiting inter-benchmark correlations enhances prediction robustness.

We also visualize the calibrated performances of all 70 LLMs among the three benchmarks (MMLU, CSQA, and Ceval) in Figure 16, with colors indicating relative ability magnitudes. This facilitates a clear comparison of LLMs' strengths on each benchmark. In particular, Figure 17 presents radar charts for three representative models, showing their overall abilities as well as benchmark-specific abilities on the same plot. Notably, gemma-3-12b-it stands out on CSQA, while gemini-2.5-pro exhibits strong performance on MMLU.

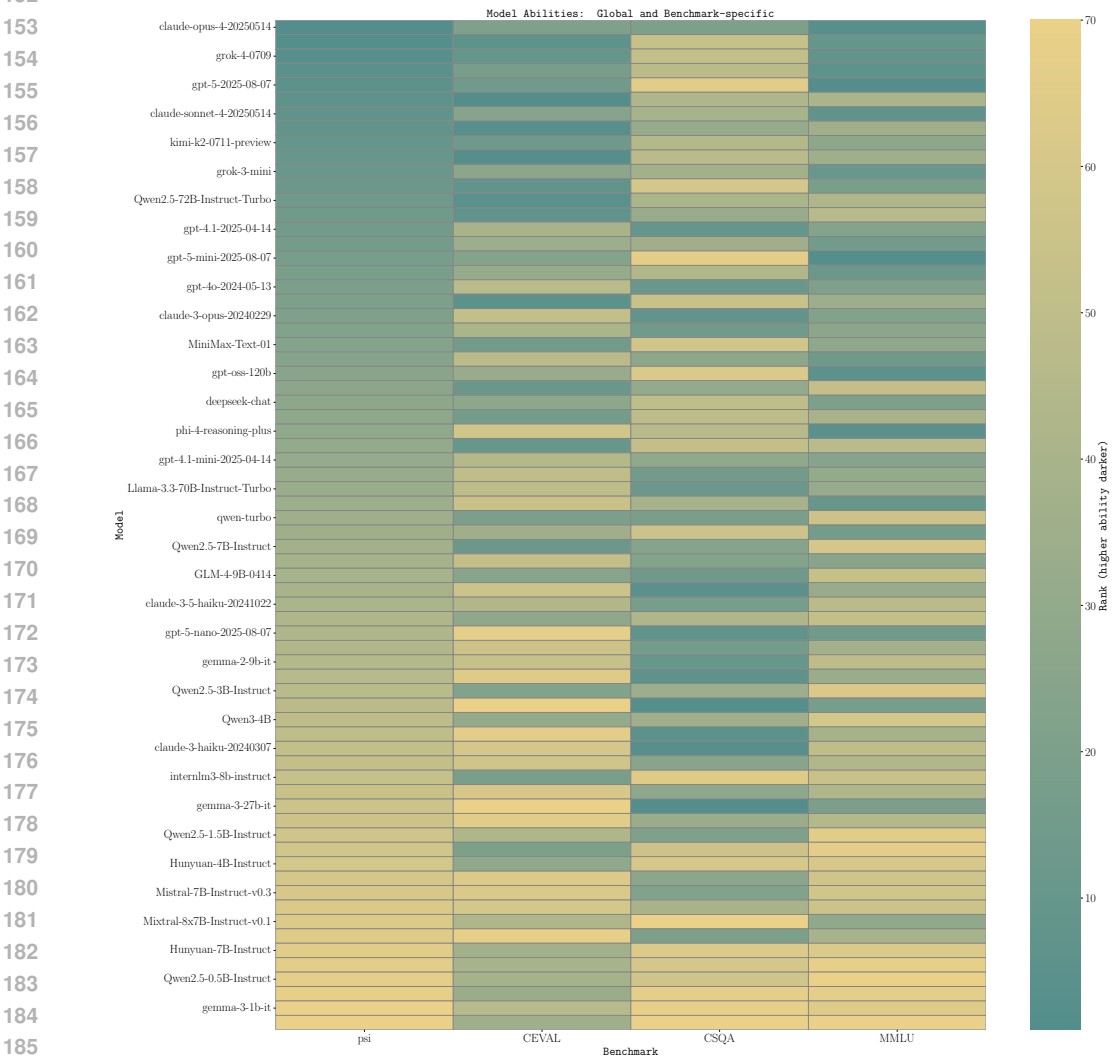

Figure 16: Estimated Model Abilities Across MMLU, CSQA, and Ceval Benchmarks

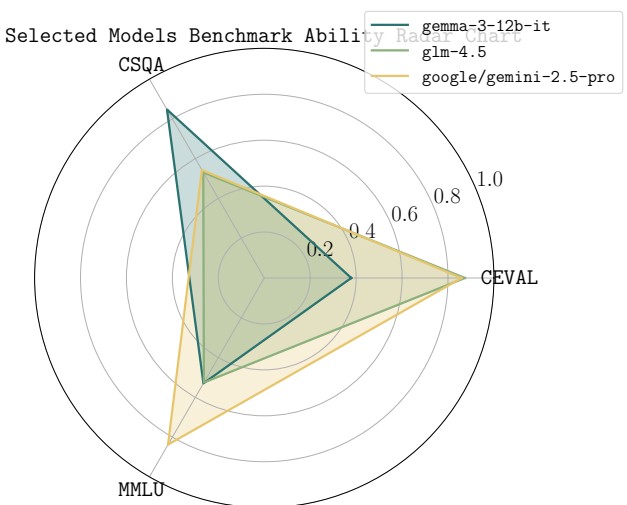

Figure 17: Benchmark-specific and Overall Model Abilities Radar Chart

Figure 18 contains two correlation matrices that illustrate the inter-benchmark relationships estimated under the mixed-benchmark IRT framework. The left matrix shows a post-hoc Pearson correlation analysis based on model accuracies on each benchmark, revealing high correlations above 0.7 among the three benchmarks. However, these correlations are in fact inflated since the native Pearson correlation fails to separate the primary factor from the secondary factors. The right matrix presents the posterior correlation matrix of benchmark-specific residuals estimated by the proposed LEGO-MB, controlling for global ability and item difficulty. Here, CSQA and MMLU show a weak positive correlation, while CEVAL and CSQA exhibit a negative correlation. Compared to the indirect post-hoc analysis, the modeled correlations better reflect the intrinsic consistency between benchmarks.

CEVAL and MMLU represent Chinese and English comprehensive language understanding benchmarks, respectively, with significant differences in language context and task content. MMLU covers diverse domains and tasks, emphasizing broad language understanding and reasoning abilities, whereas CEVAL focuses on varied tasks in Chinese contexts. The near absence of correlation between these two benchmarks indicates that model performance in one language and task environment does not directly translate to another, highlighting the complexity and challenges of cross-lingual and cross-task evaluation.

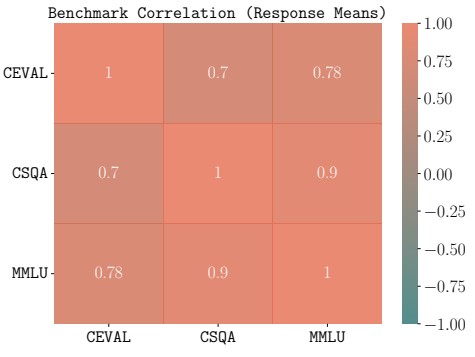
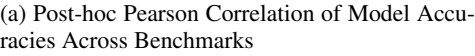

(a) Post-hoc Pearson Correlation of Model Accuracies Across Benchmarks

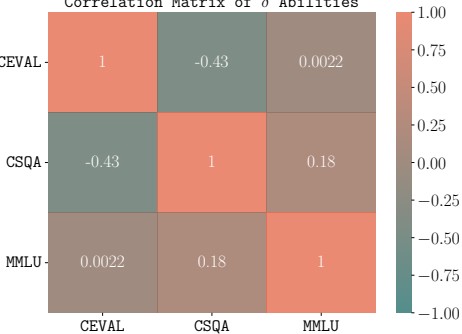

(b) Correlation Matrix of Benchmark-specific Effects Estimated by LEGO-MB

Figure 18: Inter-Benchmark Correlations: Comparison Between Post-hoc and Modeled Correlations

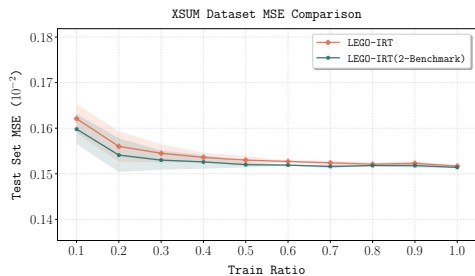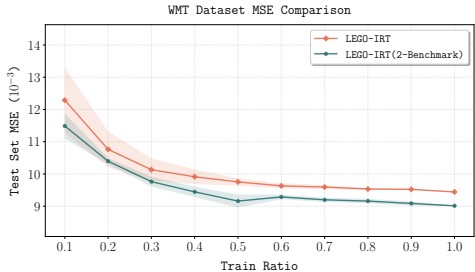

Figure 19: LLM performance prediction comparisons over XSUM and WMT20

## C.4 STRUCTURAL BENEFITS OF INTEGRATING XSUM AND WMT20

In this experiment, we use XSUM and WMT20 to assess whether the predictive performance of LEGO-IRT could be further improved over benchmarks with continuous metrics. The experimental setups are analogous to those used in the three binary benchmarks experiment. We compare the BLEU metric under the test set with MSE as the performance criterion. The results are plotted in figure 19, suggesting a significant performance gain over the WMT20 dataset.

## D DISCUSSIONS BETWEEN MCMC AND EM

Within the IRT literature, both the Expectation–Maximization (EM) algorithm and Markov Chain Monte Carlo (MCMC) methods are widely used for parameter estimation. While EM remains popular due to its computational speed, MCMC offers several methodological strengths that are particularly valuable in our LEGO-IRT framework, requiring flexible inference and robust uncertainty quantification. In this appendix, we list several advantages of MCMC over EM.

1. **Full Posterior Inference**

   EM produces point estimates by maximizing the likelihood, but it does not directly quantify uncertainty. Although we can use Louis identity (Louis, 1982) to compute the information matrix, in most cases, it can be computationally expensive when the number of parameters is large.

   By contrast, the MCMC yields full posterior distributions for item and LLMs' ability parameters, allowing for credible interval estimates, posterior predictive checks, and richer uncertainty quantification.

2. **Flexibility with Complex Models**

   EM can be difficult to extend to models with hierarchical structures, non-standard priors, or missing data patterns. In most of these cases, the E-step in EM algorithm does not admit an explicit form, which increases the computational difficulty of the M-step.

   MCMC accommodates arbitrary priors, latent variable hierarchies, and non-linear link functions, making it more suitable for our proposed LGO-IRT models with multiple metrics and benchmarks.

3. **Robustness to Multimodality**

   As we know, the log-likelihood of IRT model is not convex. EM relies on local optimization and is prone to converging at local maxima of the likelihood function.

   MCMC explores the posterior space stochastically, making it less sensitive to initialization and better at characterizing multi-modal distributions.

4. **Implicit Sparsity**

   Due to the special decomposition of latent ability under our LEGO-IRT framework, the secondary level parameters, $\zeta_{im}$ and $\delta_{i,m}$'s are only identifiable up to a location shift. In the implementation of MCMC, the priors of $\zeta_{im}, \delta_{i,m}$ implicitly push the estimates towards zero, leading to more interpretation results. On the other hand, EM algorithm does not offer this unless additional regularization terms are imposed on the $Q$-function.

