# OpenReview forum: "Toward a unified framework for data-efficient evaluation of large language models"
_ICLR.cc/2026/Conference — ICLR 2026 Conference Withdrawn Submission_

### Official Review · Reviewer_Rjwz · 2025-10-31

[review text omitted: it was posted to a different submission]

---

### Official Review · Reviewer_FzAf · 2025-11-01

**Soundness:** 2
**Presentation:** 2
**Contribution:** 2
**Rating:** 2
**Confidence:** 3

**Summary:**

The authors propose more general versions of previous IRT-based evaluations of language models. The approaches are broadly to consider more creative probabilistic graphical models and then use MCMC to perform inference. These more general probabilistic models enable extensions to continuous metrics and multiple benchmarks.

**Strengths:**

- The motivation is clear
- Extending IRT evaluations to multiple metrics and multiple benchmarks seems sensible

**Weaknesses:**

1. The paper has barely any results. The first 6 of 9 pages focus on defining different probabilistic models, and the 9th of 9th pages is a discussion, so only 2 pages (or maybe a little more) have actual results

2. The results are weak. The authors say they will focus on more modern benchmarks "We use three comprehension-type benchmarks comprising multiple choice questions (MCQ): MMLU (Hendrycks et al., 2021), CSQA (Talmor et al., 2019), and Ceval (Huang et al., 2023), among which MMLU and CSQA are English benchmarks while Ceval focuses on answering questions using Chinese." but instead focus on XSUM and WMT20. XSUM and WMT20 are odd choices given how outdated these benchmarks are; the models the authors study no longer report scores on these benchmarks. Moreover, there are many other types of benchmarks (e.g., generative mathematical and coding benchmarks like SWEBench) that this paper fails to consider.

Please see Questions.

**Questions:**

## Title

- nit: I find the title slightly non-descriptive. It’s too generic. I would recommend making it more specific e.g., “Generalizing Item Response Theory Evaluations for Large Language Model”

## Related Work

- Line 112: Why is the earliest citation for IRT from 2025? Surely there are _many_ earlier citations of IRT.

## Warmup: Item Response Theory

- Line 135: What does “local independence” mean?
- Line 141: $Y_{ij}$ should most certainly not be referred to as a metric. Accuracy is a metric. Cross entropy is a metric. If you need another word to use other than response, I would recommend calling $Y_{ij}$ a score. On line 170, the term “score” is already used :)

## Section 4 The LEGO-IRT Framework

- The motivation for the LEGO-CM probabilistic model is unclear. Many probabilistic models are probably defensible; why this one? For a naive alternative, if the goal is to parameterize a continuous distribution over [0, 1], why not use a Beta or Kumaraswamy or Continuous Bernoulli as the link distribution?

## Section 5 Experiments

- Lines 293-294: Given that MMLU, CSQA and Ceval are mentioned first, and given that these are more recent and tougher benchmarks, I’m confused why results for these benchmarks aren’t presented first? Where is the “Figure 5” variant for these benchmarks, and why is it not the actual Figure 5?

- Lines 295-296: XSUM and WMT20 are older benchmarks that I think have largely fallen out of favor for being too easy. I don’t think many of the models you evaluate even report these benchmarks. For instance, I could not find scores for these benchmarks reported in Gemini 2.5 Pro. Why were these benchmarks chosen?

- Lines 295-296: Relatedly, I think IRT approaches generally struggle when all scores are too extreme because there’s little signal to infer differential model capabilities. What were the actual models scores on XSUM and WMT20 in your study? Were they not extremely high? If not, why not? If so, did you not experience issues fitting the model parameters?

- Line 316: I like this approach whereby $r$ is swept from 0.1 to 1.0. Given what the trends look like, I feel like it may be more beneficial to try even smaller values of $r$ e.g., 0.01, and switch to a log-x scale. This is especially true for WMT Dataset MSE Comparison, since we do not know how much data is necessary for LEGO-IRT to outperform GME and MME.

- nit: Line 365 has an incorrect reference “See also figure ?? for illustration with more models”

**Details Of Ethics Concerns:**

I believe the authors modified the default ICLR 2026 template to shrink whitespace to obtain additional space for additional content that otherwise would not fit within the 9 pages limit.

To the best of my knowledge, this is not permitted under ICLR rules.

---

### Official Review · Reviewer_kspX · 2025-11-03

**Soundness:** 3
**Presentation:** 2
**Contribution:** 2
**Rating:** 4
**Confidence:** 4

**Summary:**

The paper introduces LEGO-IRT, a unified framework for data-efficient evaluation of large language models (LLMs). LEGO-IRT extends traditional Item Response Theory (IRT) by supporting both binary and continuous metrics and incorporating structural knowledge across multiple benchmarks and metrics. The framework factorizes model ability into general and structure-specific components, enabling more accurate capability estimates using only a small fraction of evaluation items. Experiments on 70 LLMs across five datasets show that LEGO-IRT provides stable and reliable estimates while reducing data requirements and estimation error.

**Strengths:**

1) The proposed LEGO-IRT extends classical IRT to support continuous evaluation metrics, multiple benchmarks and metrics.
2) The study presents a combination of theoretical formulation and empirical experiments across several datasets and quite a number of LLMs.
3) The work aims to tackle an important challenge in achieving data-efficient and generalizable evaluation for LLMs.

**Weaknesses:**

1) The evaluation benchmarks are limited to multiple-choice, summarization, and translation tasks.
2) Important LLM capabilities, such as multi-turn dialogue, complex reasoning, coding, or mathematical tasks, are not assessed, limiting task and metric diversity.
3) The link between the theoretical framework and experimental implementation is unclear. It is not explained how LEGO-IRT’s design choices are realized in the experiments.
4) The paper treats applying several metrics to the same benchmark or modeling multiple benchmarks simultaneously as “structural knowledge”, which is confusing and unclear to the reader.
5) Experimental settings are inconsistent across different sections and evaluation points, making the overall experimental logic difficult to follow.

**Questions:**

1) Could the authors clarify how LEGO-IRT would generalize to tasks beyond the evaluated benchmarks, such as multi-turn dialogue, complex reasoning, coding, or mathematical tasks?
2) How are the framework’s design choices concretely implemented in the experiments, and could the authors make the correspondence between theory and experiment clearer?
3) How are the general and structure-specific components factorized in practice, and how does this relate to the design of the experiments?
4) Could the authors clarify what is considered “structural knowledge” in LEGO-IRT, and explain why applying multiple metrics to the same benchmark or modeling multiple benchmarks simultaneously is treated as such?
5) Could you clarify why different experimental settings were used across various evaluation points, and how these inconsistencies affect the interpretability and comparability of the results?

---

### Note · Authors · 2025-12-04

I have read and agree with the venue's withdrawal policy on behalf of myself and my co-authors.